# A mechanism for CO regulation of ion channels

Sofia M. Kapetanaki[1], Mark J. Burton[2], Jaswir Basran[2], Chiasa Uragami[3], Peter C.E. Moody [2], John S. Mitcheson[2], Ralf Schmid[2], Noel W. Davies[2], Pierre Dorlet [3], Marten H. Vos [4], Nina M. Storey[2] & Emma Raven [1]

Despite being highly toxic, carbon monoxide (CO) is also an essential intracellular signalling molecule. The mechanisms of CO-dependent cell signalling are poorly defined, but are likely to involve interactions with heme proteins. One such role for CO is in ion channel regulation. Here, we examine the interaction of CO with $K_{ATP}$ channels. We find that CO activates $K_{ATP}$ channels and that heme binding to a $CXXHX_{16}H$ motif on the SUR2A receptor is required for the CO-dependent increase in channel activity. Spectroscopic and kinetic data were used to quantify the interaction of CO with the ferrous heme-SUR2A complex. The results are significant because they directly connect CO-dependent regulation to a heme-binding event on the channel. We use this information to present molecular-level insight into the dynamic processes that control the interactions of CO with a heme-regulated channel protein, and we present a structural framework for understanding the complex interplay between heme and CO in ion channel regulation.

[1] Department of Chemistry and Leicester Institute of Structural and Chemical Biology, University of Leicester, Leicester LE1 7RH, England. [2] Department of Molecular and Cell Biology and Leicester Institute of Structural and Chemical Biology, University of Leicester, Leicester LE1 9HN, England. [3] Institute for Integrative Biology of the Cell (I2BC), CEA, CNRS, Univ. Paris-Sud, Université Paris-Saclay, 91198 Gif-sur-Yvette cedex, France. [4] LOB, Ecole Polytechnique, CNRS, INSERM, Université Paris-Saclay, 91128 Palaiseau Cedex, France. These authors contributed equally: Sofia M. Kapetanaki and Mark J. Burton. Correspondence and requests for materials should be addressed to N.M.S. (email: ns140@le.ac.uk) or to E.R. (email: emma.raven@le.ac.uk)

Carbon monoxide (CO) is widely and correctly regarded as highly toxic to biological systems, by virtue of its ability to bind with high affinity to heme proteins such as haemoglobin and cytochrome $c$ oxidase. But these long-established ideas on the toxicity of CO, which originate as far back as the early experiments by J. B and J. S. Haldane[1], hide a multitude of biological roles for CO that are only very recently being unearthed. In particular, it is now known that cells actually produce CO through the enzymatic processes that are responsible for degradation of heme (catalysed by heme oxygenase[2,3]). The idea that the cellular concentration of CO, regulated in part by heme oxygenase, might itself have a wider role in cell signalling is also just emerging[4] and might include, for example, circadian and transcriptional control, gas sensing, and regulation of ion channel activity[5–8].

It has not gone unnoticed that cellular concentrations of heme and of CO may therefore be connected. From a chemical perspective, the involvement of the redox-active heme group as part of a wider mechanism for regulatory control offers considerable advantages to the cell. Diatomic gases such as CO (but also including $O_2$ and NO) are known to bind to the reduced form of heme ($Fe^{II}$). From a cell signalling perspective, and as far as overall cellular control is concerned, this offers wide versatility and chemical efficiency because the heme concentration and the $O_2$/CO concentrations are in part controlled by $O_2$-dependent and heme-dependent enzymatic processes for formation of CO (by heme oxygenase). This means that the redox state of the cell, the oxidation state of the heme, the heme concentration (i.e., the balance of heme synthesis vs heme degradation), the $O_2$ concentration, and the CO concentrations may all be inter-connected[9,10]. How this might work has not been established, but it would form a chemical basis for heme-dependent and/or CO-dependent regulatory mechanisms in the cell.

One recent example of CO-dependent regulation is in control of ion channel activity. Ion channels are central to many processes ranging from neuronal signalling to regulation of blood pressure, and thus are linked to a variety of disease states. CO-dependent regulation of channel activity has been reported in a few cases[11,12], and heme has been implicated in this CO-dependent control process, but the evidence is only empirical and there is no information on how the regulation occurs. The picture is made yet more complex by the fact that heme and CO concentrations vary in the cell[13] and are difficult to reliably quantify[14–16]. Thus, there is as yet no mechanistic picture to explain precisely how CO interacts with any ion channel, and in particular if or how this is linked to heme-binding. Because of this, the mechanisms that are involved in these processes are completely unknown at the molecular (protein) level. This is important, as CO regulation of channels has huge therapeutic implications in terms of cardiac disease, anti-inflammatory and anti-hypertensive conditions.[17]

It is therefore of significant current interest to build a more precise understanding of the role of CO in ion channel control. In this work, the aim was to use electrophysiology and a range of spectroscopies to provide insights into the interaction of CO with $K_{ATP}$ channels comprising Kir6.2/SUR2A subunits. The data provide evidence that the mechanism of CO regulation is linked to binding of heme, and provide quantification of the dynamic properties involved in the interaction of CO with heme binding domains in the channel. We use this information to present ideas on how and where CO might interact with $K_{ATP}$ channels, and draw comparisons with other heme channels that are also heme-regulated and CO-regulated.

## Results

**Heme is required for CO-dependent channel activation.** $K_{ATP}$ channels are exquisitely sensitive to the metabolic status of a cell

and subtle changes in oxidative stress affect cell excitability through alterations in $K_{ATP}$ channel activity. These channels form a hetero-octomeric structure with four Kir subunits creating the $K^+$ ion pore and four regulatory sulfonylurea receptors (SUR) subunits from the ATP-binding cassette (ABC) transporter superfamily. The most abundant cardiac $K_{ATP}$ subunits are Kir6.2 pore forming units and SUR2A receptor subunits, at the sarcolemmal membrane of cardiac muscle cells (Fig. 1a). In these cells, the production of CO is induced under oxidative stress[18]. We have demonstrated heme-dependent modulation of $K_{ATP}$ channels and we have identified a cytoplasmic region of SUR2A as the site of heme binding[19]. Gas binding proteins, such as haemoglobin, bind gasses through heme iron-based coordination. Here, we investigated the effects of CO on $K_{ATP}$ activity[20,21] and whether heme binding to the SUR2A regulatory subunit was a necessary prerequisite for CO binding to the $K_{ATP}$ channel.

To measure the effect of heme and CO on single $K_{ATP}$ channels, we transfected HEK293 cells with DNA encoding for the Kir6.2 subunit and SUR2A. Inside-out patches were superfused with heme and an increase in Popen (from $0.013 \pm 0.002$ to $0.051 \pm 0.017$, $P < 0.05$, $n = 6$, paired $T$ test) was observed. An additional application of heme with CO to the same patch resulted in a large and significant increase in channel activity ($0.303 \pm 0.064$, $^*P < 0.05$, paired $T$ test, Fig. 1b (top)). In contrast, when CO was applied alone without prior bath application of heme, no increase in channel activity was observed (Supplementary Figure 1). This suggests that heme is required for CO to cause the increase in $K_{ATP}$ Popen.

Heme binds to the $K_{ATP}$ channel at a cytoplasmic heme-binding $CXXHX_{16}H$ motif on the SUR2A subunit of the channel[19]. This $CXXHX_{16}H$ motif (residues 628–648) is located between the first transmembrane domain and the first nucleotide-binding domain, Fig. 1a (right); mutation of the $Cys^{628}$, $His^{631}$ and $His^{648}$ residues contained in this $C^{628}XXH^{631}X_{16}H^{648}$ motif removes the heme-binding ability. This triple mutation was used here in order to test whether CO interacts with the $K_{ATP}$ channel through the heme-binding region on SUR2A. $K_{ATP}$ channels containing the triple C628S/H631A/H648A mutation in the $CXXHX_{16}H$ motif were expressed in HEK293 cells and single channel activity was recorded in the same way as for the wild type channel above. In this case, the increase in $K_{ATP}$ channel activity on bath application of both heme and heme with CO was completely lost (Fig. 1b (bottom)). This indicates that binding of heme to the $CXXHX_{16}H$ motif of SUR2A is required for the CO-dependent increase in $K_{ATP}$ channel activity.

**Quantification of the heme-CO-SUR2A interaction.** To identify the molecular basis for the CO-dependent regulation of the channel observed above, a fragment of the SUR2A subunit (residues S615-L933, referred to here as $SUR2A^{615-933}$, Fig. 1a (right)) including the entire heme binding $CXXH_{16}H$ domain and the nucleotide-binding domain (NBD1) was expressed and purified (see Methods section). Absorption spectra for the heme-$SUR2A^{615-933}$ complex in the ferric ($Fe^{III}$), ferrous ($Fe^{II}$) and ferrous-CO ($Fe^{II}$-CO) forms are shown in Fig. 1c (top) (for comparison, the corresponding spectra for free heme are also shown (Fig. 1c (bottom)). The affinity of ferric heme for $SUR2A^{615-933}$ was assessed by determination of a binding constant ($K_{d,ferric} = 8.0 \pm 0.6 \mu M$, Supplementary Figure 2A) and by measuring the first-order dissociation rate constant ($k_{off} = 4.8 \pm 0.1 \times 10^{-4} s^{-1}$, Supplementary Figure 2B) for the transfer of heme from the ferric heme-$SUR2A^{615-933}$ complex to apo-myoglobin (which has a very high affinity for heme[22]). Both $K_d$ and $k_{off}$ values are in the range observed for other regulatory heme proteins, Supplementary

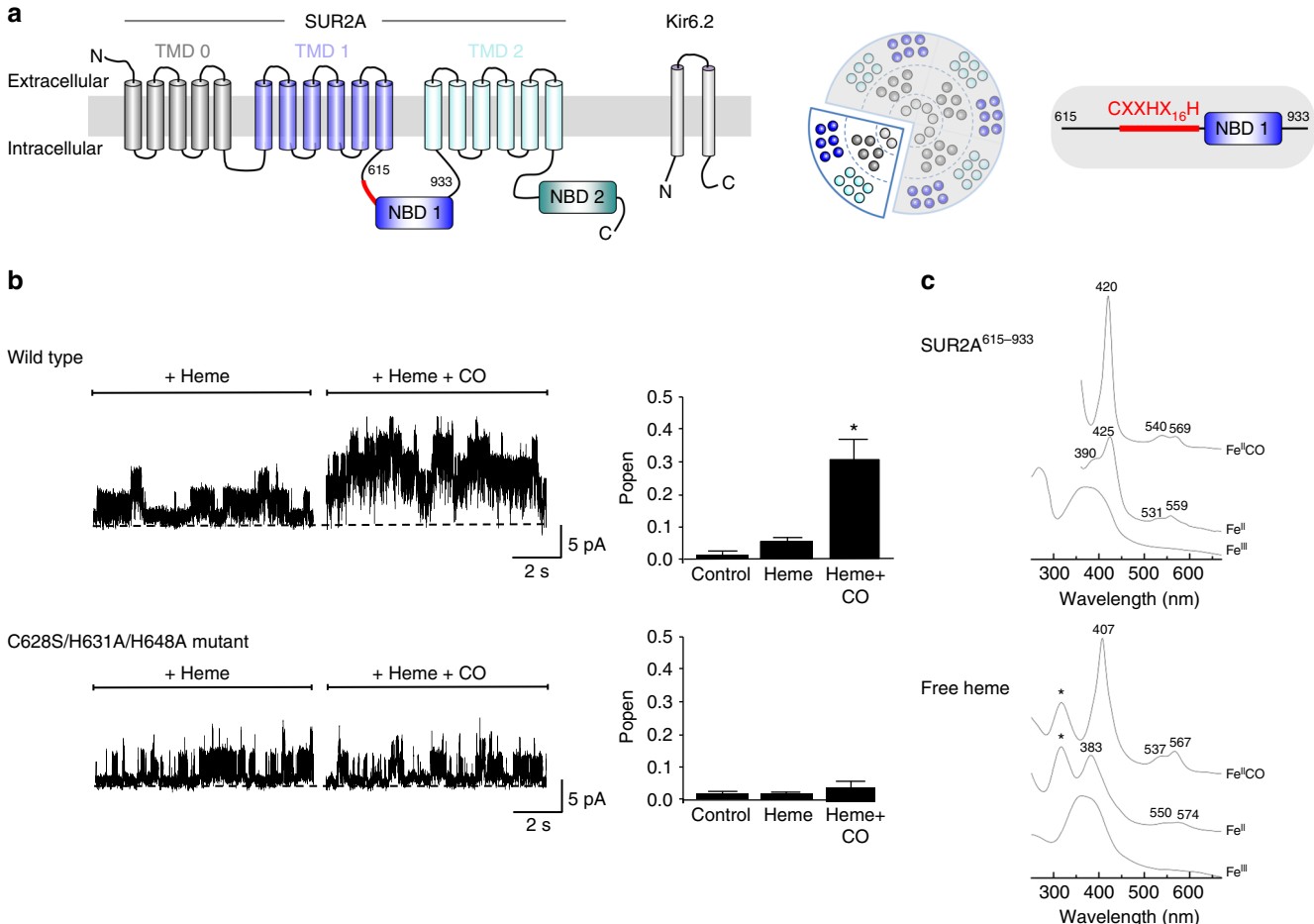

**Fig. 1** Heme is required for CO-dependent increase in channel activity. **a** Schematic of the $K_{ATP}$ channel subunit and protein structure. There are four sulphonylurea receptor (SUR) subunits which are members of the ATP-binding cassette (ABC) transporter subfamily c (SUR2A in this work), and a further four pore-lining subunits of the inward rectifier family (Kir6.2, shown on left). These assemble to form an octameric channel structure (shown in the middle). Red indicates the region of the SUR2A subunit containing the $CXXHX_{16}H$ heme binding motif adjoining the nucleotide-binding domain (NBD1). On the far right is shown schematically the S615-L933 region of rat SUR2A containing NDB1 and the $CXXHX_{16}H$ heme binding motif. **b** Mutation of residues in the heme binding region or SUR2A abolishes the heme-dependent CO increase in $K_{ATP}$ channel open probability. (Top) Inside–out single channel analysis of the effects of heme and heme/CO on wild type $K_{ATP}$ channel activity and a summary showing the mean open probability changes in response to heme and CO ($n = 6$). (Bottom) Effects of mutation of the heme binding site, in the triple C628S/H631A/H648A mutant of SUR2A, on the responses of single channel currents and mean open probability to heme and CO ($n = 6$, *$P < 0.05$, paired $T$ test). **c** Absorption spectra of SUR2A$^{615-933}$ (top three spectra) and free heme (bottom three spectra) in the ferric (Fe$^{III}$), ferrous (Fe$^{II}$) and ferrous CO-bound (Fe$^{II}$-CO) forms. The ferric form of free heme is characterized by a broad Soret band centered at 377 nm[53], which probably arises from different heme conformations; reduction of free heme with dithionite leads to a red-shifted Soret band at 383 nm which is attributed to a 5-coordinate high spin species and closely resembles that of dithionite-reduced hemin in 0.5% sodium dodecyl sulphate[54] ($\lambda_{max}$ = 395, 405, 425 (weak), 439 (weak) nm). Addition of CO to reduced heme results in the formation of a complex with a Soret band at 407 nm and Q-bands at 537 and 657 nm[55]

Table 1, and are several orders of magnitude higher than those of the globins which bind heme much more tightly.

The ferrous derivative of the heme-SUR2A$^{615-933}$ complex has a split Soret band at 425 nm with a shoulder at 390 nm, and Q-bands at 531 and 559 nm (Fig. 1c (top)). The 425 nm species is characteristic of 6-coordinate, low-spin ferrous heme[23]. Addition of CO to the ferrous heme-SUR2A$^{615-933}$ complex leads to the formation of a stable heme-CO species, with a sharp Soret band ($\lambda_{max}$ = 420 nm) and Q bands at 540 and 569 nm (Fig. 1c (top)). This spectrum is markedly different to that seen for the ferrous-CO complex in free heme (compare Fig. 1c (bottom)) and is closely analogous to the spectrum of the heme-CO complexes observed in myoglobin and in a number of His/Cys-ligated regulatory heme proteins in which CO displaces the Cys ligand (Supplementary Table 2). The spectrum also differs from the CO complex in P450, which are also Cys-ligated ($\lambda_{max} \approx 440$ nm[24]).

Supplementary Table 2 presents a comparison of absorption maxima for these proteins.

We went on to quantify in greater detail the heme-CO-SUR2A$^{615-933}$ binding interaction. The affinity of CO for the ferrous heme-SUR2A$^{615-933}$ complex is in the low μM range ($K_{d, CO}$ = 0.6 ± 0.3 μM, Fig. 2a), which is consistent with the known binding affinities for other regulatory heme proteins (Supplementary Table 3). The ferrous heme-SUR2A$^{615-933}$ complex is also competent for binding of other closely related diatomic gases (Supplementary Figure 2c), as would be expected for a bona fide heme protein.

The dynamics of binding of CO to the ferrous heme-SUR2A$^{615-933}$ complex are similar to other 6-coordinate heme proteins (e.g. [25]). The fastest phase of the binding event is dependent on [CO] (Fig. 2b), and shows a non-linear dependence on the concentration of CO. This is consistent with a mechanism

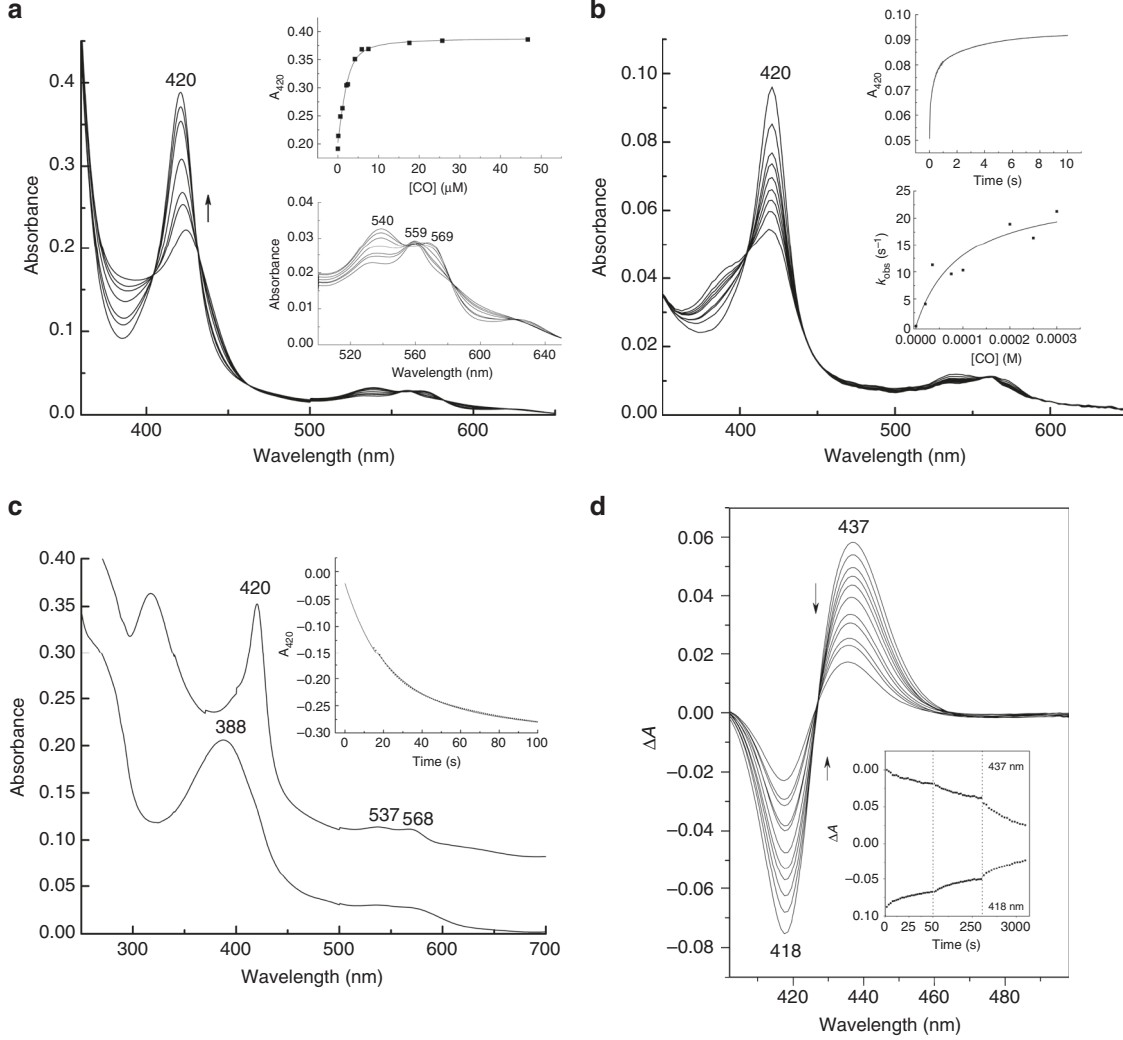

**Fig. 2** CO binds to heme-binding motif on SUR2A. **a** Spectrophotometric titration of the ferrous heme-SUR2A[615–933] complex with CO. Inset top: quadratic (Morrison) fitting to the binding curve to give $K_d = 0.6 \pm 0.3$ μM; inset bottom: an expansion of the visible region. **b** Stopped-flow spectra (pH 7.5, 22.0 °C) observed on binding of CO (300 μM) to the ferrous heme-SUR2A[615–933] complex (2 μM). Inset top: kinetics transient observed on binding of CO (300 μM) to the ferrous heme-SUR2A complex, monitored as an increase at 420 nm; data were fitted to a three-exponential process. Inset bottom: Observed rate constants for the first phase, $k_{obs}$ (s$^{-1}$), plotted as a function of CO concentration; there is an asymptotic phase with a rate constant ($k_1$) that corresponds to the off-rate of the distal residue. A fit of $k_{obs}$ to the hyperbolic equation [1] (see Results section) gives $k_1 = 25 \pm 5$ s$^{-1}$ and an effective second order rate constant at low [CO] $k_{CO} = k_1k_2/k_{-1} = 0.27 \pm 0.10$ μM$^{-1}$s$^{-1}$. Using $k_{-2} = 0.05 \pm 0.01$ s$^{-1}$ (as calculated from ligand substitution, Fig. 2c), a value for $K_d = k_{-2}/k_{CO} = 0.19 \pm 0.10$ μM is predicted which is in good agreement with the $K_d$ measured independently ($K_d = 0.6 \pm 0.3$ μM, Fig. 2a). **c** Kinetics and spectral changes associated with dissociation of CO from the ferrous heme-SUR2A[615–933] complex (3 μM) in the presence of NO. NO was produced by the NO releasing molecule, S-nitroso-N-acetyl penicillamine (SNAP) (200 μM). Absorption spectra show the reactant (the heme-CO-SUR2A[615–933] complex, top spectrum), and the product (the heme-NO-SUR2A[615–933] complex, bottom spectrum). The inset shows the 420 nm time trace, along with a three-exponential fit; using the dominant (~ 70%) phase a rate constant for dissociation of CO, $k_{-2}$, was determined ($k_{-2} = 0.05 \pm 0.01$ s$^{-1}$). **d** Transient absorption spectra of the heme-CO-SUR2A[615–933] complex at various delay times after photodissociation of the CO ligand (10, 30, 60, 100, 150, 300, 700, 900, 1700, 2100 and 3500 ps). Inset shows transients at 437 nm and 418 nm observed for the heme-CO-SUR2A[615–933] complex

in which there is a conformational rearrangement of the heme-SUR2A[615–933] complex prior to the ligand binding event, as shown in Equation 1, where SUR2A[6c] represents a 6-coordinate species, SUR2A[5c] represents a 5-coordinate species that can bind CO, and SUR2A-CO represents the heme-CO-SUR2A[615–933] complex.

$$\text{SUR2A}^{6c} \underset{k_{-1}}{\overset{k_1}{\rightleftharpoons}} \text{SUR2A}^{5c} \underset{-\text{CO},k_{-2}}{\overset{+\text{CO},k_2}{\rightleftharpoons}} \text{SUR2A} - \text{CO} \qquad (1)$$

For $k_{-2} << k_1k_2/k_{-1}$ this equation yields a hyperbolic dependence on CO concentration of the observed CO binding rate $k_{obs}$ of the

form

$$k_{obs} = k_1k_2[\text{CO}]/(k_{-1} + k_2[\text{CO}]) \qquad (2)$$

A fit of the data to Eq. 2 (Fig. 2b), yields a value for the limiting first-order rate constant for dissociation of the distal ligand ($k_1$ in Eq. 1) of $25 \pm 5$ s$^{-1}$, which is in the range expected when compared with other heme proteins which also undergo conformational rearrangement from 6-coordinate to 5-coordinate on ligand binding, Supplementary Table 3. A composite second order rate constant, $k_{CO} = 0.27 \pm 0.10$ μM$^{-1}$ s$^{-1}$, that reports on CO binding can be derived from this non-linear data

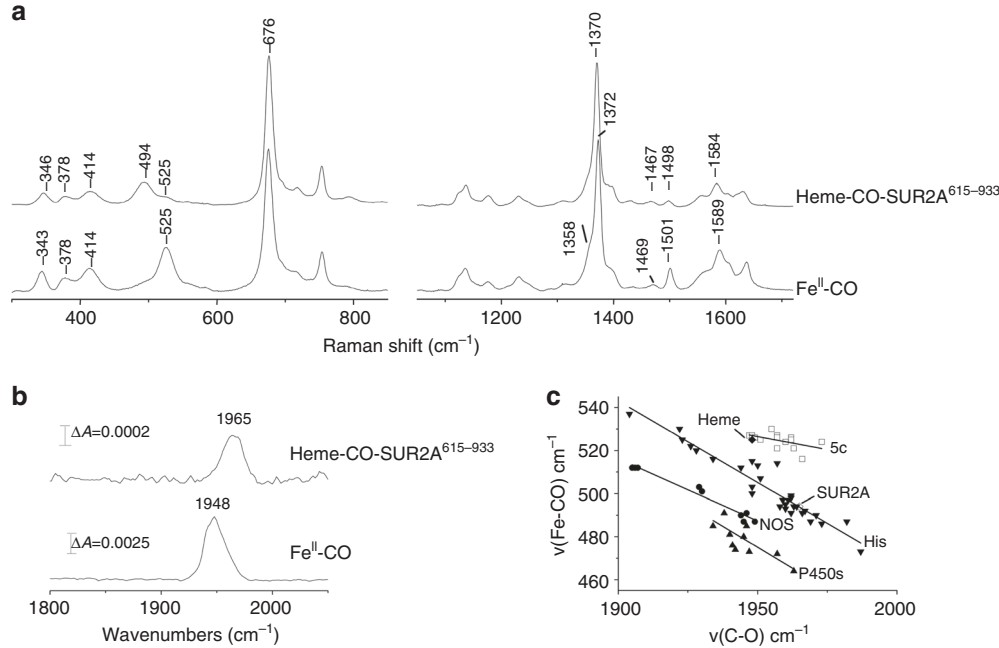

**Fig. 3** Resonance Raman spectroscopy identifies the nature of the heme-CO-SUR2A complex. **a** Resonance Raman spectra in the high- and low-frequency of the CO-bound complexes of ferrous-heme and ferrous heme-SUR2A[615–933], obtained with 413.1 nm excitation. **b** ATR (attenuated total reflection) infrared spectra of the CO-bound complexes of ferrous-heme and heme-SUR2A[615–933] in $D_2O$ in the 1750–2050 $cm^{-1}$ region. **c** A plot of ν(Fe–CO) and ν (C–O) stretching frequencies for the CO complexes of various heme proteins and model compounds with different axial ligation. All entries in this plot were created from the data presented in Supplementary Table 5; data obtained for model compounds are shown as squares, for histidine-ligated heme proteins are shown as inverted triangles, for nitric oxide synthases are shown as circles, and for P450s are shown as triangles. SUR2A[615–933] (indicated as unfilled diamonds) and free heme (indicated as diamonds) lie on the correlation line that is characteristic for histidine-ligated heme proteins (inverted triangle) and 5-coordinate model compounds with no *trans* ligands or 6-coordinate heme species with weak trans ligands (square), respectively

(see Methods section); this value is in the range expected when compared with other heme proteins, Supplementary Table 3. With the independently determined off-rate constant ($k_{-2}0.05\,s^{-1}$, Fig. 2c), a $K_d$ value for CO binding ($K_d = k_{-2}/k_{CO} = 0.19 \pm 0.10\,\mu M$) can also be extracted from the data; this value for $K_d$, determined kinetically, is in reasonably close agreement with that determined by ligand titration ($K_d = 0.60 \pm 0.30\,\mu M$, Fig. 2a).

**Resonance Raman identifies the heme-CO-SUR2A interaction**. To further probe the interactions of CO with heme in the channels, we used resonance Raman spectroscopy. Resonance Raman can provide rich and sensitive information on the precise nature of protein-heme-CO interactions—in part because CO, which is a π-acid ligand, competes for the iron $d_\pi$ electrons, giving rise to upfield shifts of π-sensitive porphyrin marker bands. This makes CO an excellent probe of heme environment because it is very sensitive to electrostatic and steric interactions[26,27]. In CO-bound complexes of heme proteins the key vibrational modes are the ν(Fe–CO) stretching mode (460–540 $cm^{-1}$) and the weak ν (C–O) stretching modes (1900–1990 $cm^{-1}$); the δ(Fe–C–O) bending modes (550–570 $cm^{-1}$) are often weak and not observed in protein-free heme-CO complexes[28].

High-frequency and low-frequency resonance Raman spectra of the ferrous-CO complex of free heme are shown in Fig. 3a. The high-frequency region (1300–1700 $cm^{-1}$) comprises porphyrin in-plane vibrational modes that are sensitive to the electron density in the porphyrin macrocycle ($v_4$) and to the coordination and spin state of central iron ($v_3$, $v_2$)[29]. Upon binding of CO to ferrous heme, the $v_4$, $v_3$ and $v_2$ bands are observed at 1372 $cm^{-1}$, 1501 $cm^{-1}$ and 1589 $cm^{-1}$ respectively, arising from a ferrous

heme-CO adduct with a weak proximal ligand (probably water)[30]. A band at 1469 $cm^{-1}$ and a shoulder at 1358 $cm^{-1}$ suggest that partial photodissociation of CO from the heme takes place. In the low-frequency region, the band at 525 $cm^{-1}$ is assigned to a ν (Fe–CO) stretching vibration. The ν(C–O) mode is at 1948 $cm^{-1}$ (Fig. 3b). A δ(Fe–C–O) bending mode (550–570 $cm^{-1}$ region) is not observed, which suggests a nearly linear Fe–C–O geometry, as expected in this (free) heme-CO complex.

The corresponding high-frequency resonance Raman spectrum of the heme-CO-SUR2A[615–933] complex (Fig. 3a), is similar to that of the heme-CO complex, with slight differences in the intensity of the $v_3$ band (at 1498 $cm^{-1}$) and in the 1610–1640 $cm^{-1}$ region where the $v_{10}$ and vinyl stretching modes of the heme are observed. A comparison of the vibrational modes observed for the heme-CO-SUR2A[615–933] complex with other known heme proteins is provided in Supplementary Table 4, and demonstrates similarities in the spectra. In the low-frequency region, the heme-CO-SUR2A[615–933] complex displays a dominant ν(Fe–CO) stretching vibration at 494 $cm^{-1}$ and a weak vibration at 525 $cm^{-1}$. In the high-frequency region, a dominant Fe–CO stretching vibration at 1965 $cm^{-1}$ is observed with a shoulder at 1948 $cm^{-1}$ (Fig. 3b). Deconvolution of the 450–600 $cm^{-1}$ and 1850–2050 $cm^{-1}$ region with Gaussian line shapes (width 16–24 $cm^{-1}$) indicates that ≈85% of the heme-CO is bound to protein. A nearly linear Fe–C–O geometry is suggested for the CO-bound complex of SUR2A[615–933] as no intense δ(Fe–C–O) bending mode is present in the 550–570 $cm^{-1}$ region.

Analysis of the ν(C–O) and ν(Fe–C) stretching frequencies is informative to further quantify the interaction of CO within the heme-CO-SUR2A[615–933] complex. Back donation of the dπ electrons of the iron to the π* orbitals of the CO ligand increases the Fe–C bond strength while at the same time decreasing the

C–O bond strength; this leads to a negative linear correlation between the $\nu$(C–O) and $\nu$(Fe–C) stretching frequencies, which are affected by the polarity of the heme environment and the identity of the ligand *trans* to CO[31]. A comparative plot of $\nu$(Fe–CO) vs $\nu$(CO) stretching frequencies observed in SUR2A and in numerous other heme proteins is presented in Fig. 3c; in this plot, the heme-CO-SUR2A$^{615-933}$ complex lies on a correlation line that is characteristic of heme proteins containing a proximal histidine ligand and at a position on the line that is indicative of diminished polarization and hence of a hydrophobic environment around the bound CO.

**Dynamics of the heme-CO interaction in SUR2A.** Ultrafast recombination of gaseous ligands with heme is a very sensitive probe of heme environment[32]. We therefore used ultrafast spectroscopy to gain insight into the dynamics of CO binding to the heme-SUR2A$^{615-933}$ complex.

Selected transient spectra observed following photodissociation of CO from the heme-CO-SUR2A$^{615-933}$ complex are shown in Fig. 2d. A red shift is observed that is typical for transitions from six-coordinate (i.e., a His-ligated ferrous-CO complex) to five-coordinate (i.e., a His-ligated ferrous complex) species on loss of CO. After spectral relaxations due to photophysical processes (over time scales of a few picoseconds[32]), the spectra become almost identical in shape but with diminished amplitudes, reflecting substantial geminate heme-CO recombination on the picosecond- to early nanosecond time scale. This rebinding of CO to heme was found to be highly efficient (~90% see below), indicating that the heme environment of SUR2A$^{615-933}$ acts as an effective ligand trap. The kinetics of rebinding span at least 2 orders of magnitude in time (Fig. 2d (inset)), showing three exponential phases of 22 ps (14%), 150 ps (26%) and 2.5 ns (50%) and a long-lived phase (10%). Multiphasic geminate rebinding behaviour is observed in many heme proteins[32] but mostly not over such large time spans (Supplementary Table 6). This large span of rebinding rates is likely to reflect a wide distribution of either enthalpic barriers reflecting different configurations of the dissociated heme and its environment (as assigned by Champion and coworkers in very recent work on the temperature dependence of heme-CO rebinding in CooA[33] and in free heme in solution[34]) and/or of configurations of dissociated CO (in particular their orientations) in the heme pocket[35]. Heme binding to a flexible region of protein between TMD1 and NBD1 (Fig. 1a, left), would account for this wide distribution of configurations in the SUR2A$^{615-933}$ complex (see Discussion section).

**Discussion**

That CO might act as a beneficial signalling molecule in cells, rather than being simply toxic, overturns century-old ideas on the role of CO in biology. Several ion channels are reported as targets for CO[11,36] and heme is implicated in a number of cases, but the study of CO signalling in ion channels is in its infancy. Binding of CO (which is a strong $\pi$-acceptor ligand) to a heme-containing protein offers a chemically plausible mechanism for the regulation, but the mechanisms of regulation are yet to be established and often contradictory[12]. Understanding these events at a molecular level is important as it underpins progress on therapeutic interventions, but decoupling these various ion channel control mechanisms is hugely challenging and requires more than an empirical demonstration of CO response in a particular ion channel.

In this paper we have, first of all, demonstrated that CO regulates $K_{ATP}$ channels and that activation of the channel depends on the presence of heme. We have also established that CO binds tightly to the (ferrous) heme-SUR2A$^{615-933}$ complex, and that the

binding interactions are analogous to those in other well-established heme systems. This is significant because it directly connects a CO regulatory event to a heme binding process on the channel. Our data for the heme-SUR2A$^{615-933}$ complex are consistent with a 6-coordinate low-spin heme species with His and Cys as the heme axial ligands, as shown schematically in Fig. 4. On addition of CO, the Cys ligand is displaced to give a CO-bound complex with a histidine as proximal ligand (Fig. 4). Fe-Cys bonds are typically weak, and even weaker in the ferrous (Fe$^{II}$-Cys) than in the ferric (Fe$^{III}$-Cys) forms, and so are expected to dissociate in the presence of a strong $\pi$-acceptor ligand such as CO[37]. The affinity of ferric heme for SUR2A$^{615-933}$ (in the $\mu$M range) is consistent with a flexible and potentially reversible (i.e., non-covalent) heme binding interaction that would accommodate a regulatory role for heme and CO in the channel. By way of comparison, most well-known heme proteins—such as the *b*-type cytochromes and the globins—also bind heme non-covalently but the interactions are essentially irreversible by virtue of strong coordinate bond(s) to axial heme ligands (usually His, Met, or Cys) within a tightly defined heme pocket; because of this, these proteins thus have much higher affinities for heme, usually in the pM[38] or nM range[39,40]. In contrast, the interaction of heme with regulatory/trafficking proteins appears to be different—less rigid, more flexible, and therefore weaker. This gives different spectroscopic signatures from "traditional" heme proteins. In the case of the $K_{ATP}$ channels, our data suggest a conformationally mobile heme pocket, which might have a role to play in cell signalling as changes in protein structure on binding of heme or CO can potentially be transmitted to other regions of the subunit structure.

A chemical basis for heme-dependent regulation and the connections that might exist between heme and CO in the channel is outlined schematically in Fig. 4. The biosynthesis of heme (itself an 8-step process) in part regulates the overall supply. Of the total heme concentration, some is committed to the "housekeeping" heme proteins (cytochrome *c*, globins, peroxidases, cytochrome *c* oxidase etc.) essential for the survival of the cell. The rest of the heme—sometimes referred to as labile heme[15,16]— is thought to exist as "pool" and to be available for regulatory control. The concentrations of heme in this pool are difficult to quantify but estimates are in the low $\mu$M to nM range[15,16,41]. It is not known in what form heme exists in this pool, but is likely bound to so-far unidentified transporter proteins (although the transport mechanisms are also largely unknown). Some of this heme is presumed to be available to bind to ion channel proteins found at the plasma membrane (Fig. 4). Under conditions of excess heme (e.g., during hemeorrhage or thrombosis), upregulation of the $O_2$-dependent heme oxygenase enzyme occurs, which depletes heme concentrations (by degradation of heme) and leads to formation of CO which can feed back into regulatory control of ion channels by interacting with heme bound to channel proteins.

There is no structural information available for heme binding to an ion channel, but recent cryo-EM structures[42,43] of ATP-sensitive potassium channels help us to piece together a structural framework for the interpretation of our data, as they reveal the binding orientations of SUR1 and Kir6.2 subunits. There are three isoforms of the SUR subunits—SUR1 (from pancreas), and two splice variants of SUR2 (SUR2A from cardiac muscle as in this work, and SUR2B in smooth muscle, see Fig. 5a). SUR1 is the closest homologue (67% identity) to SUR2A in the ABC subfamily *c* but the heme binding CXXHX$_{16}$H motif, in the linker region between TMD1 and NBD1 of SUR2A, is missing in SUR1 (Fig. 5a). Both cryo-EM structures[42,43] suggest that this linker region forms a flexible or disordered region as it is not resolved in the structures, but the location of the Kir6.2, NBD1 and TMD1

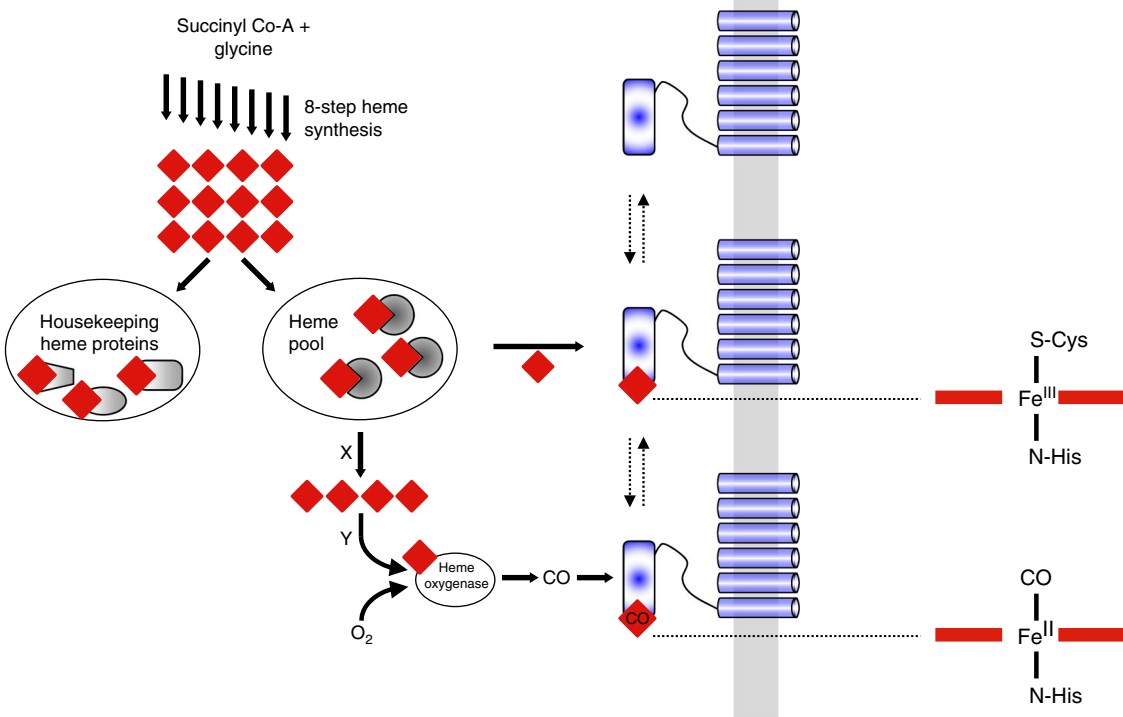

**Fig. 4** A molecular basis for CO-dependent activation of $K_{ATP}$ channels. Figure (left) shows the connections that might exist between heme (red diamond) and CO. The 8-step biosynthesis of heme regulates the overall supply. Of the total heme concentration, some is committed to the heme proteins essential for the survival of the cell (left oval). The remainder is thought to exist in a 'pool' (right oval), probably bound to unknown chaperone proteins. Heme in this pool is available for regulatory control, including binding to ion channel proteins (blue) bound at the membrane (grey). Partner proteins (X, Y) are envisaged as important for heme transport and/or for reduction of the heme to the ferrous ($Fe^{II}$) state (prior to binding of gaseous ligands). Formation of CO from the catalytic degradation of heme by the $O_2$-dependent heme oxygenase enzyme is envisaged to also interact with heme bound to channel proteins (see text for details). On the far right is shown the proposed ligation in the heme-SUR2A$^{615-933}$ and heme-CO-SUR2A$^{615-933}$ complexes. The Cys/His ligation may be retained in the ferrous form, but $Fe^{II}$-thiolate bonds are weak and replacement by a second (protein) ligand is also possible[37, 56]. Note that the activating effects of CO on the channel are not observed in the absence of heme or when the heme binding site on SUR2A is removed

domains indicates its approximate position (Fig. 5b). In this position, the heme binding region is accessible from the cytoplasm where changes in heme concentration could be detected. Conformational change upon binding of heme and CO could thus be transmitted to the adjacent Kir6.2, thereby providing a mechanism to control channel gating. We have not directly assessed the potential physiological consequences of such a change in conformation, but increased Kir6.2 and SUR2A channel activity could conceivably alter the excitability of tissues that express these $K_{ATP}$ channels (such as cardiac muscle).

Might other ion channels, that are also regulated by heme and CO, behave similarly? This has yet to be established, but the cryo-EM structure of the high-conductance $Ca^{2+}$-activated $K^+$ channel[44] (also known as Slo1), which is also heme-regulated[45] and CO-regulated[12], has identified the location of a cytochrome c-type CXXCH heme-binding motif that has been implicated[45] in heme-dependent regulation of the Slo1 channel. Comparison of the locations of the CXXCH motif in Slo1 (Fig. 5c) with the CXXHX$_{16}$H region of Kir6.2/SUR2A (compared Fig. 5b,c) highlights common features. In both $K_{ATP}$ and Slo1 channels, the heme binding regions are located within the intracellular domains of the channel, and are therefore accessible. If, as our kinetic data indicate, the process of heme/CO binding is also accompanied by a conformational change in protein structure that leads to the replacement of the Cys ligand by CO, then one can envisage other, more global conformational changes in channel structure (for example around the linker region between SUR2A and the

pore) that are induced by this local heme/CO binding event and that might therefore affect channel opening/closing. If this is the case, it may be more than just coincidence that the heme binding domains are located close to unstructured regions in both $K_{ATP}$ and Slo1 channels, as a conformational change to a more structured and less dynamic state would provide an additional layer of channel gating control.

## Methods

**Chemicals and reagents**. Aqueous stock solutions of hemin for spectroscopic and electrophysiological experiments were prepared by dissolving solid hemin in 0.1 M NaOH. Final concentration of stocks were calculated spectrophotometrically at 385 nm ($\varepsilon_{385} = 58.4$ mM$^{-1}$ cm$^{-1}$)[46] and diluted accordingly to 500 nM prior to use. All reagents were from Sigma-Aldrich (Dorset, England) unless otherwise stated. CO solutions were prepared by bubbling bath solution with CO gas.

**Transfection and cell culture**. (Human embryonic kidney 293 (HEK293) cells were cultured on sterile glass coverslips in MEM (Life Technologies) with 10% FBS, 1% L-glutamine, and 1% streptomycin/penicillin and incubated at 37 °C, 5% $CO_2$ for at least 24 h before transfection or patch-clamp recordings were made. Transfection of HEK293 cells with plasmids encoding recombinant Kir6.2 and SUR2A was carried out using Lipofectamine 2000 (Life Technologies) 12–24 h before recording. The rat Kir6.2-pIRES-eGFP and SUR2A-pCMV subunits were a generous gift from David Lodwick, University of Leicester, Leicester, United Kingdom.

**Electrophysiology**. Single channel currents (filtered at 2 kHz) were recorded for 2 min at pipette potential of +70 mV in inside-out patches with an Axopatch 200B and sampled at 10 kHz with a Digidata 1440A. The test potential of −70 mV gave a high driving force at a voltage where other channels are not activated. The conductance of the recorded channels was ~78 pS under our conditions which is

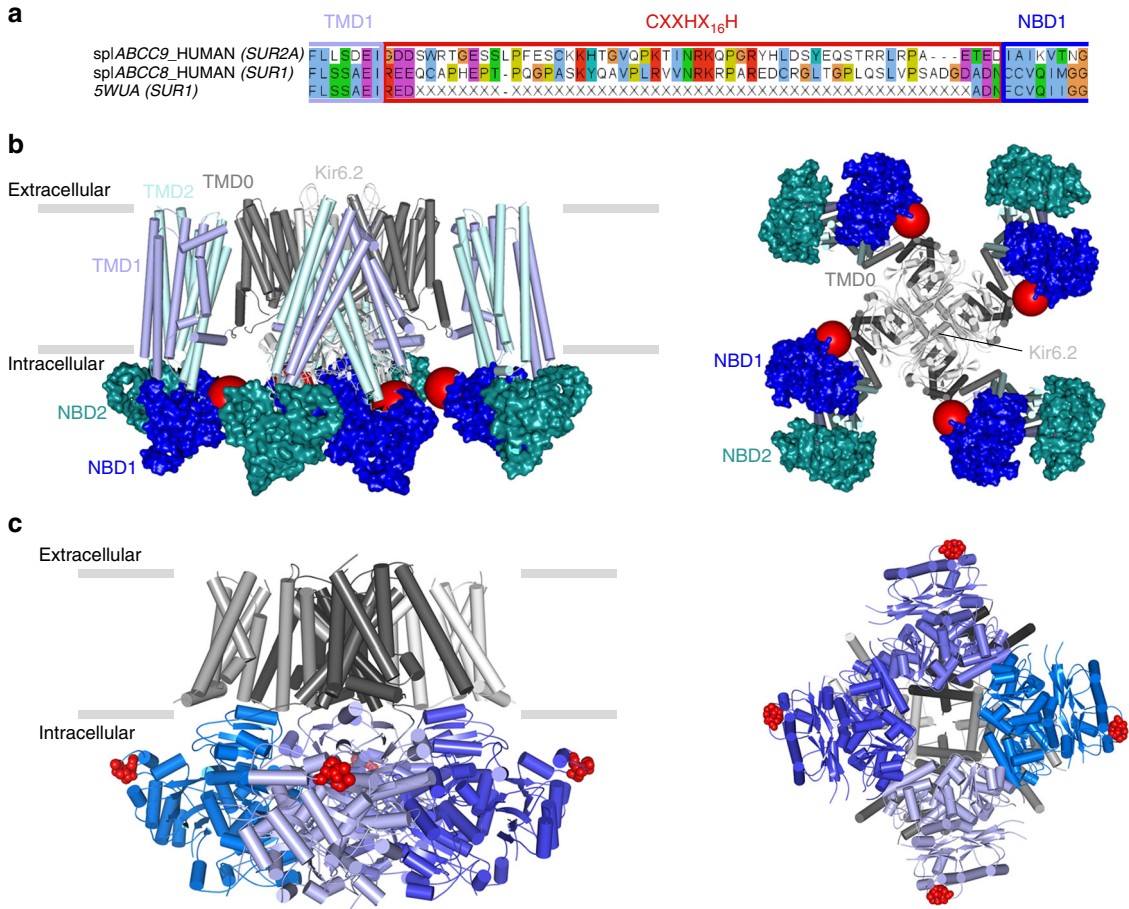

**Fig. 5** A structural basis for heme binding. **a** Sequence alignment between two isoforms of the SUR subunit—SUR2A (ABCC9_HUMAN) and SUR1 (ABCC8_HUMAN and 5WUA). Only the heme binding region of SUR2A and the equivalent region of SUR1 (red box) are shown; the adjacent TMD1 and NBD1 domains are indicated by boxes in light and dark blue respectively. Residues not resolved in the sequence of the equivalent region of the 5WUA cryo-EM structure[42] are listed as 'X'; the clustal colour scheme indicates biophysical properties of amino acids. **b** Domain structure of the Kir6.2/SUR2A $K_{ATP}$ channel mapped on the EM structure of the pancreatic ATP-sensitive potassium channel (PDB 5WUA[42]), viewed along the membrane plane (left) and perpendicular to the membrane from the intracellular space (right). The colour scheme follows that in Fig. 1: Kir6.2 (light grey), TMD0 (dark grey), TMD1 (light blue) and TMD2 (light cyan) domains of SUR2A in cartoon representation; NBD1 (dark blue) and NBD2 (dark cyan) of SUR2A are shown in surface representation. The region corresponding to the heme binding domain in SUR2A is not resolved in the corresponding SUR1 domain of 5WUA, its putative location is indicated by a red sphere. **c** Domain structure of the Slo1 channel tetramer (PDB 5TJ6[44]) shown in an equivalent orientation to (**b**) above. Transmembrane domains are in grey, intracellular RCK1/2 domains in blue, and individual monomers are indicated by different shades of grey/blue, respectively. The CXXCH heme binding motif in Slo1 is resolved in the structure (in red). Unlike Kir6.2/SUR2A, where the heme binding region is part of an unstructured region which is not resolved in the structure (red sphere in (**c**)), the heme binding motif in Slo1 is resolved (actual residues shown in red) but in close proximity to other unstructured residues/regions which are not visible in the structure

consistent with Kir6.2 and SUR2A channel recordings. The number of channels per patch was variable with an average of $5.4 \pm 0.3$ channels ($n = 12$). Patch pipettes (8–10 M$\Omega$) were pulled from thick-walled borosilicate glass and filled with solution comprising (mM): 140 KCl, 1.2 MgCl$_2$, 2.6 CaCl$_2$, 5 HEPES, pH 7.4. Bath solution contained (mM): 30 KOH, 110 KCl, 10 EGTA, 1.2 MgCl$_2$, 1 CaCl$_2$, 5 HEPES, 1 reduced glutathione, pH 7.2. ATP (500 μM) was applied to the inside-out patches in all recordings with heme. All CO solutions were prepared by bubbling bath solution with CO gas; this creates a saturated CO solution ([CO] = 1mM[47]). All experiments were conducted at room temperature ($22 \pm 1$ °C). Data are presented as the mean ± SEM of $n$ experiments and * denotes statistical significance $P < 0.05$ paired $T$ test.

**Expression and purification of SUR2A NBD1.** Residues S615-L933 of rat SUR2A (referred in this paper as SUR2A$^{615–933}$) were expressed and purified following standard procedures[19]. Briefly, E. coli cell cultures were grown and protein expression was induced by addition of IPTG (isopropyl β-D-1-thiogalactopyranoside). Cells were harvested, resuspended and loaded onto a Ni$^{2+}$-NTA (Qiagen) affinity column. The His$_6$-SUMO tag from the eluted protein was cleaved with a His-tagged TEV-protease and SUR2A NBD1 was purified to homogeneity with size exclusion chromatography. Purity of the final preparations was assessed by SDS/PAGE. Protein concentration was determined by Bradford protein assay.

**Optical absorption spectroscopy.** Absorption spectra were obtained using a double beam spectrophotometer (Perkin Elmer Lambda 40) or a Kontron Uvikon UV–vis spectrometer. Hemin was added to SUR2A$^{615–933}$ in 50 mM phosphate, 50 mM NaCl pH 8.00 or 50 mM HEPES, 50 mM NaCl, pH = 7.5 and spectra were recorded after hemin addition. The corresponding reduced forms were obtained by addition of dithionite in anaerobic samples; ferrous heme bound to SUR2A is stable for hours under anaerobic conditions. Ferrous-CO complexes were formed by direct bubbling of CO into pre-reduced samples generated by addition of a slight excess of dithionite to ferric complexes.

**Resonance Raman spectroscopy.** Samples (50 μL) of ferrous heme, ferrous-CO heme and their complexes with SUR2A$^{615–933}$ in 50 mM HEPES, 50 mM NaCl pH 7.5 at 90–120 μM were prepared in quartz EPR tubes and disposed in a homemade spinning cell, at room temperature, to avoid local heating and to prevent photo-dissociation and degradation. Raman excitation at 413.1 nm was achieved with a laser power of ~1 mW for the ferrous-CO samples. Resonance Raman spectra were recorded using a modified single-stage spectrometer (Jobin-Yvon T64000, HORIBA Jobin Yvon S.A.S., Chilly Mazarin, France) (spectral resolution 3 cm$^{-1}$) equipped with a liquid N$_2$-cooled back-thinned CCD detector. Stray scattered light was rejected using a holographic notch filter (Kaiser Optical Systems, Ann Arbor, MI). Spectra correspond to the average of three different 1 h accumulations. The spectral accuracy was estimated to be ±1 cm$^{-1}$. Baseline correction was performed

using GRAMS 32 (Galactic Industries, Salem, NH) and Origin®. Sample integrity was verified by following RR spectral evolution during the experiment.

**ATR-FTIR measurements**. FT-IR spectra were recorded with a Nicolet 6700 FTIR spectrometer using OMNIC software. The spectrometer was equipped with a DTGS CsI detector and an XT-KBr beam-splitter. Samples were prepared in deuterated buffer at a heme concentration 500 μM and placed on the ATR Si prism (3 mm diameter, three bounce, SensIR Europe) and covered to avoid oxidation by air. Each spectrum is the average of 1024 scans and the spectral resolution was 4 cm$^{-1}$.

**EPR spectroscopy**. EPR spectra were recorded on an Elexsys 500 X-band spectrometer (Bruker) equipped with a continuous-flow ESR 900 cryostat and an ITC504 temperature controller (Oxford Instruments, Abingdon, UK). Experimental conditions: microwave frequency 9.38 GHz, microwave power 4 μW, field modulation frequency 100 kHz, field modulation amplitude 0.5 mT, $T$ 10 K. Simulations were performed by using the Easyspin software package[48] and routines written in the lab.

**Determination of heme affinity**. Optical titrations to determine the $K_d$ values for the various ligands (hemin, CO) were performed at 25.0 °C using SUR2A$^{615-933}$ at ~3–5 μM in 50 mM HEPES/50mM NaCl, pH 7.5. Stocks solution of ligand were prepared in the same buffer and added stepwise to the sample cuvette containing SUR2A$^{615-933}$. Spectra were recorded after each addition. In the case of hemin binding, hemin was also added to the reference cuvette containing the same buffer and difference absorption spectra were recorded[19]. Values of $K_d$ were determined by fitting to a hyperbolic equation or the Morrison (quadratic) equation (for a tight-binding ligand, CO) (OriginLab, Northampton, MA).

**Kinetic studies**. Pre-steady state stopped-flow experiments were carried out using an Applied Photophysics SX.20MV stopped-flow spectrometer housed in an anaerobic glove box (Belle Technology Ltd.; [O$_2$] < 5 ppm) and fitted with a Neslab RTE-200 circulating water bath (25.0 ± 0.1 °C). In stopped-flow experiments, stated concentrations of protein and reagents relate to final concentrations in the flow (after mixing).

The kinetics of heme dissociation from the ferric heme-SUR2A$^{615-933}$ complex were measured spectrophotometrically using a Perkin Elmer Lambda 40 spectrophotometer. Transfer of hemin to myoglobin on reaction of the ferric heme-SUR2A$^{615-933}$ complex (6.4 μM) with a 5.6-fold excess of apo-myoglobin (36 μM) was followed at 408 nm (experiments were carried out in 50 mM HEPES/ 50 mM NaCl pH 7.5).

Rate constants for binding of CO to the ferrous heme-SUR2A$^{615-933}$ complex were determined by stopped-flow. Solutions of ferrous heme-SUR2A$^{615-933}$ (2.0 μM) in 50 mM HEPES/50 mM NaCl, pH 7.5 were rapidly mixed with an equal volume of CO buffer (at varying CO concentrations, in excess over protein concentration) and the absorbance changes in the 350–650 nm region were monitored. Buffers at differing concentrations of CO were prepared by appropriate dilution of the stock saturated CO solution ([CO] = 1 mM[47]) with anaerobic (oxygen-free) buffer. Data were analysed by fitting to the appropriate rate equations supplied with the Pro-K software package (Applied Photophysics). Nonlinear dependencies on [CO] were observed, according to the mechanism shown in Eq. 1 (see Results section) and fitted to the corresponding Eq. [1], $k_{obs} = k_1k_2[CO]/(k_{-1} + k_2[CO])$. Values for the limiting first-order rate constant, $k_1$, as well as an overall composite second-order rate constant for CO binding, $k_{CO} = k_1k_2/k_{-1}$, can be extracted from the data at low concentrations of [CO] (as derived in ref.[25, 49]).

The dissociation rate constant, $k_{-2}$, for loss of CO from the ferrous heme-SUR2A$^{615-933}$ complex was determined by ligand replacement[50]. Typically, a small volume (approximately 20 μl) of 9 mM S-nitroso-N-acetyl penicillamine was transferred to an anaerobic cuvette containing the heme-CO-SUR2A$^{615-933}$ complex (900 μl, 3 μM). Release of CO was monitored by the rate of disappearance of the absorbance at 420 nm. Approximately 75% of a heme-NO-SUR2A$^{615-933}$ complex forms with a rate in the order of 0.05 s$^{-1}$.

**Ultrafast spectroscopy**. Proteins were suspended at a concentration of ~50 μM in 50 mM HEPES/50 mM NaCl buffer, pH 7.5. Experiments were performed in 1 mm path length optical cells sealed with a gastight stopper. Oxygen was removed by several cycles of vacuum pumping and exposure to pure argon gas and the gas phase was replaced by 100% CO or 10% NO. The samples were left to equilibrate until formation of the Fe$^{II}$-CO and Fe$^{II}$-NO complexes were complete, as monitored by the steady-state absorption spectrum.

Multicolour femtosecond absorption experiments were performed on a 500 Hz repetition rate setup as described[51], based on a Quantronix Integra-C Ti-Saph oscillator/amplifier system, with a pump pulse centred at 570 nm and a broad band continuum probe pulse extending down to ~350 nm generated in a continuously translated CaF$_2$ window. Both test and reference probe beams pass through the sample, which is rastered with a Lissajous scanner. Data were globally analyzed in terms of multi-exponential decay using the Glotaran package[52].

**Data availability**. Data supporting the findings of this manuscript are available from the corresponding authors upon reasonable request.

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

## Acknowledgements

We thank C. Owen and D. Roper for providing the pSUMODAVE vector; X. Yang for preparing the SUR2A expression clone; A. Pascal, M. J. Llansola, M. Weisslocker-Schätzel and A. Lukács for technical assistance; W. Leibl for access to FT-IR; B. Robert for access to Raman equipment, J. Santolini for assistance with data collection, and U. Liebl for helpful discussions. This work was funded by Biotechnology and Biological Sciences Research Council Grants BB/K000128/1 and BB/M018598/1. We acknowledge the French Infrastructure for Integrated Structural Biology Grant ANR-10-INSB-05-01.

## Author contributions

S.M.K., M.J.B., R.S., P.C.E.M., M.H.V., P.D., J.S.M., N.W.D., N.M.S., and E.L.R. designed the research. S.M.K., M.J.B., J.B., C.U., P.D., and M.H.V. performed experiments. S.M.K., M.J.B., J.B., P.D., M.H.V., R. S., N.M.S. and E.L.R. analysed data; S.M.K., M.J.B., N.M.S, and E.L.R wrote the paper with contributions from all authors.

## Additional information

**Competing interests:** The authors declare no competing interests.

