## [Peer Review File · Nature Communications]

Editorial Note: This manuscript has been previously reviewed at another journal that is not operating a transparent peer review scheme. This document only contains reviewer comments and rebuttal letters for versions considered at Nature Communications. Mentions of prior referee reports have been redacted.

Reviewers' comments:

Reviewer #1 (Remarks to the Author):

This is a very interesting paper that I think should be published in Nature Communications. However, I do think there is some level of speculation or confusion surrounding the necessity of CO binding to heme as the prime regulator of these ion channels (see point 9 below). A more direct discussion of how the present study interfaces with the author's prior work in ref.19 (where just the heme seems to be regulating the ion channel) would be helpful. In this context, the statements on page 2 (2nd paragraph of Results) should be made more quantitative. The phrasing: "increase" in P_{open} vs "large and significant increase" (when CO is present) could be revised to include some numbers consistent with Fig. 1B(I), which should help to clarify the author's message. I have a few other comments and questions that should be answered prior to publication.

1) On page 2 the mutation studies are described near the bottom of the page. It is not discussed in this paper why these three residues were chosen for the mutation studies. Why all 3 at once? What happens when residues are mutated one at a time? How were these 3 residues selected for mutation in the first place? Probably there is a secondary discussion of this in ref 19, but a short comment here would be helpful to the casual reader.

2) On the top of page 3, I would again suggest using statements that are less general than "markedly higher". Something a bit more specific or quantitative might be "many orders of magnitude higher".

3) I found Eq. 1 to be rather confusing, especially in the context of the ensuing discussion concerning k_{CO} , k_{on} , and k_{off} . The latter two rates must be defined in terms of the quantities given in Eq. 1. However, Eq. 1 appears to be incomplete because it does not explicitly include the CO binding rate that leads to the formation of the heme-CO product "P". This equation and the ensuing discussion must be fixed prior to publication.

4) At the end of the paragraph where Eq. 1 is presented, the K_{d} values are compared for kinetics and equilibrium studies. I do not understand why the authors claim "close agreement" when these values appear to differ by a factor of three. There is almost an overlap at the extremes of the error bars, but I think the phrase describing this comparison could be made more appropriate. I would also ask the authors to comment on why the agreement is not better.

5) At the bottom of page 4 there is a claim that "the CO samples a wide distribution of heme environments and/or CO configurations with respect to the heme within the protein structure prior to rebinding". The latter possibility is generally thought to lead to a distribution of entropic barriers, which enter the rate expression as part of a temperature-independent prefactor, whereas the former, when it involves heme conformations, involves a distribution of enthalpic barriers. Recently published work on CooA (J. Am. Chem. Soc., 2017, 139, 15738) reveals that this ubiquitous non-exponential behavior is strongly temperature dependent. Thus, the non-exponential effects appear to be primarily due to a distribution of enthalpic rebinding barriers related to heterogeneity in the heme configurations. This result holds for a number of other heme-CO temperature studies (e.g., including the free heme-CO as discussed in PNAS 2007, 104, 14682). The phrasing at the bottom of page 4 should be appropriately modified to reference these prior assignments of temperature dependent non-exponential kinetic response related to heme conformational heterogeneity.

6) In the 2nd paragraph of the Discussion, there is a comment that the Cys ligand is expected to dissociate in the presence of a strong π -acceptor ligand such as CO. A more complete explanation of this effect would be useful, but if there is not enough room for a full explanation, a good reference to

why this might be taking place would be helpful to the reader.

7) At the end of the main paper, just prior to the Methods section there is a statement: "If, as our kinetic data indicate, protein conformational changes accompany the heme/CO binding event...". It is not clear what conformational changes are being referred to in this sentence. Is it the ligand switching process where the Cys ligand is lost, or is it some other conformational change? It would be helpful to clarify what aspect of the kinetic data indicates this and what type of conformational change is taking place.

8) The absorption spectrum of ferric SUR2A in Figure 1.c.I seems different from what is displayed in Fig. S2. The former has a broad Soret band that resembles the ferric free heme; whereas the spectra in Fig. S2 depicts a sharp Soret band with a maximum located at ~414 nm. Which spectrum is the correct one?

9) Figure 4 caption part (a) at the end there is a confusing suggestion: "..., suggesting that CO regulation need not be limited only to heme-binding locations." Is this statement just a typo that infers the opposite of what the authors actually mean? My reading of the manuscript was that the heme was required for CO regulation (end of the second paragraph of Results). If so, the regulation must be limited to the where the heme binds, which seems to contradict the phrasing at the end of the Fig. 4(a) caption.

Reviewer #2 (Remarks to the Author):

There remain gaping holes in our understanding of CO as a signaling agent. Questions regarding the regulation of heme oxygenase activity to produce CO in response to a cellular signal and unequivocally established receptors for CO have yet to be reported. Indeed, there are stress responses that lead to HO-1 expression but the tie to a specific CO response has eluded investigators. Complicating matters is the potential for non-physiological CO concentrations to interfere with NO signaling, thus leading to discussions of CO responses.

This paper is focused on ion channels as a target for CO and follows up on previous work, including more importantly a 2016 PNAS paper by this research group (ref 19). This current submission builds on results in that paper with significant overlap. For example, use of the SUR2A peptide was developed in the PNAS paper. Oddly, the PNAS paper reports spectra of various complexes but only in the ferric oxidation state of the heme. For CO to bind, the heme would need to be in ferrous oxidation state. This current submission does extend the spectroscopic studies to ferrous complexes. In that sense, this paper is certainly an advance. Heme affinity measurements and potential iron ligands is also a new and important part of this paper. Many of the cellular ion channel experiments build on and extend their previous results.

It is clear that heme and CO influence ion channel activity. The central question remains and that is one of physiological relevance. The model as described would have heme in the ferric oxidation state "delivered" to the heme binding site in the channel. Although not discussed, at some point reduction to the ferrous oxidation would have to take place. Then the stage would be set for CO binding. All experiments here with ferrous heme were carried out under anaerobic conditions. What is the stability of the of the ferrous bound heme? What is the redox potential of the ferric-ferrous couple? Without data on these aspects, this next step in the story is not as significant as it could be (or should be).

Reviewer #3 (Remarks to the Author):

The manuscript by Kapetanaki et al. reports CO modulation of the KATP channel via interaction with the heme binding domain present on the SUR2A subunit of the channel complex. It provides functional and chemical analytic evidence for a direct protein-protein interaction via a conserve heme binding domain. The manuscript is well written and the experimental strategies are appropriate, however there are some issues that should be addressed before publication can be recommended. These issues are outlined below and can be rectified. Therefore my recommendation is publication on revision:

- There is a question over the originality of the manuscript. Several studies have demonstrated the role of CO in modulation of ion channels, ranging from functional to biochemical studies (e.g. Williams et al., 2004; Jaggar et al., 2005). This has included the investigations examining direct interactions with the channel protein (Jaggar et al., 2005; Williams et al., 2008; Brazier et al., 2009). Pertinent to this study others have indicated a CO and Katp interaction (Foresti et al., 2004; de Avila et al., 2014).
- There is a lack of clarity over the relative CO concentrations used in the experiments outlined. For example; for the electrophysiology experiments, what concentration of CO do the authors conclude is acting on the channel protein? Is a concentration response present? This could be clarified by spectrophotometric measurement of carbonmonoxymyoglobin.
- The physiological role of this interaction has not been demonstrated, the studies have focused on overexpression of channel protein. How do the authors perceive that this interaction occurs at a system level?
- Mutational work (Figure 1b); are all residues required for the observed CO effect?
- Functional datasets (Figure 1b); single channel K+ conductance plots should be presented. Why did the authors select +70mV as a test potential? Did the authors estimate the number of channels per patch? What was the sample size for the data presented?

Minor Comments

P2. 'Inside-out patches were superfused with heme and an increase in Popen was observed.' The data does not support this statement and it should be revised.

Reviewer 1

This is a very interesting paper that I think should be published in Nature Communications. However, I do think there is some level of speculation or confusion surrounding the necessity of CO binding to heme as the prime regulator of these ion channels (see point 9 below). A more direct discussion of how the present study interfaces with the author's prior work in ref. 19 (where just the heme seems to be regulating the ion channel) would be helpful.....

We have included a more expansive comment at the beginning of the results (section "Binding of heme is required for CO-dependent channel activation"). We hope this is acceptable.

..... In this context, the statements on page 2 (2nd paragraph of Results) should be made more quantitative. The phrasing: "increase" in P_{open} vs "large and significant increase" (when CO is present) could be revised to include some numbers consistent with Fig. 1B(I), which should help to clarify the author's message.

Agreed. This paragraph on p2 now includes quantitative data taken from the graphs numbers in Fig1B, as requested.

I have a few other comments and questions that should be answered prior to publication.

1) On page 2 the mutation studies are described near the bottom of the page. It is not discussed in this paper why these three residues were chosen for the mutation studies. Why all 3 at once? What happens when residues are mutated one at a time? How were these 3 residues selected for mutation in the first place? Probably there is a secondary discussion of this in ref 19, but a short comment here would be helpful to the casual reader.

Agreed. We have amended the text on p2 to give a more complete explanation.

2) On the top of page 3, I would again suggest using statements that are less general than "markedly higher". Something a bit more specific or quantitative might be "many orders of magnitude higher".

Agreed. We have adjusted the text accordingly, it now reads "Both K_d and k_{off} values are in the range observed for other regulatory heme proteins, **Table S1**, and are several orders of magnitude higher than those of the globins which bind heme much more tightly." Hope this is acceptable.

3) I found Eq. 1 to be rather confusing, especially in the context of the ensuing discussion concerning k_{CO} , k_{on} , and k_{off} . The latter two rates must be defined in terms of the quantities given in Eq. 1. However, Eq. 1 appears to be incomplete because it does not explicitly include the CO binding rate that leads to the formation of the heme-CO product "P". This equation and the ensuing discussion must be fixed prior to publication.

We agree it is quite complicated! It arises because the mechanism has more than one rate-limiting step so that the observed overall (on) rate constant for CO binding (k_{obs}) is a composite rate constant that is defined by the individual microscopic rate constants. For clarity, we have now renamed Eq 1 Scheme 1, and explicitly added the ensuing equation to which the CO binding data are fitted as Eq. 1. For low [CO], $k_{CO} = k_1 k_2 / k_{-1}$ (we have given that definition in the legend to Figure 2 and the methods). Hence, Scheme 1 is correct but does not explicitly include the experimental rate k_{CO} .

To further help simplify, we have removed all reference to k_{on} (now referred to only as k_{CO} , the second order rate constant deducible from the observed CO binding rate k_{obs} at low [CO]), we have replaced k_{off} and $k_{off(CO)}$ with k_{-2} , and this simplifies everything (!) because now we can replace the expression $K_d =$

$k_{\text{off}}/k_{\text{on}}$ with $K_{\text{d}} = k_{-2}/k_{\text{CO}}$. We have kept the terminology of $k_{\text{on}} / k_{\text{off}}$ in relation to heme dissociation because Olson has used that in the past and it might be confusing to use $k_{\text{-heme}}$ and $k_{\text{+heme}}$ instead. We have changed the supplementary methods to make everything consistent. We have also added k_{-2} to Eq 1 and changed P in Eq 1 to make it clear that we mean the SUR2A-heme-CO complex. We have also changed the slightly awkward $k_{\text{off,distal}}$ in Table S3 to k_1 , which now fits with Eq 1.

4) *At the end of the paragraph where Eq. 1 is presented, the K_{d} values are compared for kinetics and equilibrium studies. I do not understand why the authors claim “close agreement” when these values appear to differ by a factor of three. There is almost an overlap at the extremes of the error bars, but I think the phrase describing this comparison could be made more appropriate. I would also ask the authors to comment on why the agreement is not better.*

A three-fold difference in K_{d} is very close agreement, and we consider those values to be essentially the same, particularly since the values are obtained from different kinetic ($K_{\text{d}} = 0.19 \pm 0.1 \mu\text{M}$) and equilibrium ($K_{\text{d}} = 0.6 \pm 0.3 \mu\text{M}$) methods. In terms of why the constants are not in closer agreement, the reason they are not identical is that unfortunately both values come with sizeable errors - the largest relative error, from the titration curve, arises from the fact that the protein concentration required to accurately perform the measurements lies above the K_{d} .

We have looked again at the text on p3, and it does explain the kinetic/equilibrium measurements. Bearing in mind the readership of this journal, we don't consider that an adjustment of the p3 text is helpful as it will over-complicate the issue and disrupt the flow of text. The legend for Figure 2B, which we have referred to, is already quite detailed on this point and we think it is best to leave that detail of how the calculation is made to the legend. But please advise if the reviewer thinks we need to make a further change.

5) *At the bottom of page 4 there is a claim that “the CO samples a wide distribution of heme environments and/or CO configurations with respect to the heme within the protein structure prior to rebinding”. The latter possibility is generally thought to lead to a distribution of entropic barriers, which enter the rate expression as part of a temperature-independent prefactor, whereas the former, when it involves heme conformations, involves a distribution of enthalpic barriers. Recently published work on CooA (J. Am. Chem. Soc., 2017, 139, 15738) reveals that this ubiquitous non-exponential behavior is strongly temperature dependent. Thus, the non-exponential effects appear to be primarily due to a distribution of enthalpic rebinding barriers related to heterogeneity in the heme configurations. This result holds for a number of other heme-CO temperature studies (e.g., including the free heme-CO as discussed in PNAS 2007, 104, 14682). The phrasing at the bottom of page 4 should be appropriately modified to reference these prior assignments of temperature dependent non-exponential kinetic response related to heme conformational heterogeneity.*

The reviewer refers to very recent work of the Champion group that was published after the submission of the present manuscript. In this work, the temperature dependence of the heterogeneous heme-CO recombination in the bacterial CO sensor CooA was interpreted in terms of a temperature dependent distribution of heme configurations leading to a temperature-dependent distribution of enthalpic rebinding barriers. This interpretation contrasts with recent work involving one of us (M.H.V.) on the non-related bacterial CO sensor RcoM2, in which arguments were put forward to explain the (quantitatively quite different) kinetic heterogeneity in terms of protein motions induced CO configurational heterogeneity. As both interpretations require a very flexible protein environment, which is the main point we want to make in this context, and as we do not consider this paper the right forum for a detailed discussion of this physical-chemical issue, we have now rephrased the text at this point so as to explicitly mention both possibilities referring to the corresponding work. We hope the reviewer sees this as a sensible balance.

6) *In the 2nd paragraph of the Discussion, there is a comment that the Cys ligand is expected to dissociate in the presence of a strong π -acceptor ligand such as CO. A more complete explanation of this effect would be useful, but if there is not enough room for a full explanation, a good reference to why this might be taking place would be helpful to the reader.*

Agreed. The heme-SUR2A complex is a type-2 iron heme center (Burstyn, Chemical Reviews 2015, 115, 2532). Type-2 thiolates are known to be replaced by a redox-mediated ligand switch or by the CO molecule itself, which we had mentioned in the legend to Figure 4B. Rather than get bogged down with a full explanation in the Discussion, we think a reference is best here, so we have included the recent Burstyn review. We have also adjusted the legend to Figure 4A, to include the Burstyn reference.

7) *At the end of the main paper, just prior to the Methods section there is a statement: “If, as our kinetic data indicate, protein conformational changes accompany the heme/CO binding event...”. It is not clear what conformational changes are being referred to in this sentence. Is it the ligand switching process*

where the Cys ligand is lost, or is it some other conformational change? It would be helpful to clarify what aspect of the kinetic data indicates this and what type of conformational change is taking place.

Agreed. We spent quite a long time worrying about this sentence. In an earlier version we had been more expansive about what kind of conformational changes we had in mind, but we removed it because we only had evidence for some of these conformational changes. Since it is the last part of the paper, we were trying to strike a balance between what we have seen (evidence for conformational changes from the kinetic data) into other conformational changes that might affect channel gating but are purely speculative on our part. We have adjusted the text accordingly at the end of the Discussion, hopefully that now looks better.

8) The absorption spectrum of ferric SUR2A in Figure 1.c.I seems different from what is displayed in Fig. S2. The former has a broad Soret band that resembles the ferric free heme; whereas the spectra in Fig. S2 depicts a sharp Soret band with a maximum located at ~414 nm. Which spectrum is the correct one?

Both spectra are correct. The one in Fig1C(I) is an absolute spectrum of SUR2A-hemin while the one in Fig. S2a is a difference spectrum (after free hemin has been subtracted). These are already marked as such in the legends, but please advise if further clarification is needed.

9) Figure 4 caption part (a) at the end there is a confusing suggestion: "..., suggesting that CO regulation need not be limited only to heme-binding locations." Is this statement just a typo that infers the opposite of what the authors actually mean? My reading of the manuscript was that the heme was required for CO regulation (end of the second paragraph of Results). If so, the regulation must be limited to the where the heme binds, which seems to contradict the phrasing at the end of the Fig. 4(a) caption.

Agreed. We have removed this part of the sentence.

Reviewer 2

There remain gaping holes in our understanding of CO as a signaling agent. Questions regarding the regulation of heme oxygenase activity to produce CO in response to a cellular signal and unequivocally established receptors for CO have yet to be reported. Indeed, there are stress responses that lead to HO-1 expression but the tie to a specific CO response has eluded investigators. Complicating matters is the potential for non-physiological CO concentrations to interfere with NO signaling, thus leading to discussions of CO responses.

Thank you for the comments; no response needed.

This paper is focused on ion channels as a target for CO and follows up on previous work, including more importantly a 2016 PNAS paper by this research group (ref 19). This current submission builds on results in that paper with significant overlap. For example, use of the SUR2A peptide was developed in the PNAS paper. Oddly, the PNAS paper reports spectra of various complexes but only in the ferric oxidation state of the heme. For CO to bind, the heme would need to be in ferrous oxidation state. This current submission does extend the spectroscopic studies to ferrous complexes. In that sense, this paper is certainly an advance. Heme affinity measurements and potential iron ligands is also a new and important part of this paper. Many of the cellular ion channel experiments build on and extend their previous results.

Thank you for the comments; no response needed.

It is clear that heme and CO influence ion channel activity. The central question remains and that is one of physiological relevance. The model as described would have heme in the ferric oxidation state "delivered" to the heme binding site in the channel. Although not discussed, at some point reduction to the ferrous oxidation would have to take place....

Agreed – it was implicit in our discussion although we did not state it very well. The best place to make this clearer is in the legend to figure 4A, which is now adjusted accordingly.

.....Then the stage would be set for CO binding. All experiments here with ferrous heme were carried out under anaerobic conditions. What is the stability of the of the ferrous bound heme? What is the redox potential of the ferric-ferrous couple? Without data on these aspects, this next step in the story is not as significant as it could be (or should be).

Ferrous heme bound to SUR2A is stable for hours under anaerobic conditions. We have inserted a statement on p11 about that.

We agree that the measurement of the reduction potential would be a useful value, but this is not a straightforward measurement in these difficult ion channel proteins. Using a method that we have published previously (Efimov et al. FEBS Letters, 2014, 588, 701-704) we have estimated the potential of SUR2A to be around -200 mV, but we did not include these data because the Efimov method (which uses redox active dyes) seems to partially destabilise the SUR2A during the experiment. If the potential of

SUR2A is low, it would actually fit with ideas about how heme regulatory proteins are stored in cells – it is suggested that they would be more stable in the ferric form and would only be reduced as and when needed (as discussed for CoxA by Aono in *Acc. Chem. Res.* 2003, 36, 825-831, or by Cowley *et al* in *Inorg. Chem.* 2006, 45, 9985-10001). Our decision in preparing the paper for publication was that it was too speculative to include in the final version.

Reviewer 3

The manuscript by Kapetanaki et al. reports CO modulation of the KATP channel via interaction with the heme binding domain present on the SUR2A subunit of the channel complex. It provides functional and chemical analytic evidence for a direct protein-protein interaction via a conserve heme binding domain. The manuscript is well written and the experimental strategies are appropriate, however there are some issues that should be addressed before publication can be recommended. These issues are outlined below and can be rectified. Therefore my recommendation is publication on revision.

Thank you for the supportive comments.

There is a question over the originality of the manuscript. Several studies have demonstrated the role of CO in modulation of ion channels, ranging from functional to biochemical studies (e.g. Williams et al., 2004; Jaggar et al., 2005). This has included the investigations examining direct interactions with the channel protein (Jaggar et al., 2005; Williams et al., 2008; Brazier et al., 2009).....

We are of course aware of those papers. We had cited the Williams 2004 paper, but in discussing the effects of CO we had chosen to direct the reader to excellent review articles (Peers 2015, Wilkinson 2011, Peers, 2011, Peers 2012) rather than citing individual papers which provide only empirical evidence for a CO effect (e.g. Jaggar 2005, Williams 2008). We agree with the reviewer that the Jaggar, 2005 and Williams, 2008 papers, both on Slo1, have examined direct interactions with the channel – these papers are certainly very helpful contributions, but they are limited in their structural interpretations because they do not quantify or analyse the interaction with the channel (and the Williams paper only uses electrophysiology). Our paper goes much further than this, by providing a fully quantitative analysis of the CO-binding interactions *and* connecting them with a heme binding event.

..... Pertinent to this study others have indicated a CO and Katp interaction (Foresti et al., 2004; de Avila et al., 2014).

As above, the Foresti and de Avila papers only present empirical demonstrations of a heme/CO effect on KATP channels, but without any evidence for *how* that process occurs. We have cited these papers, as they are relevant to this particular channel.

There is a lack of clarity over the relative CO concentrations used in the experiments outlined. For example; for the electrophysiology experiments, what concentration of CO do the authors conclude is acting on the channel protein? Is a concentration response present? This could be clarified by spectrophotometric measurement of carbonmonoxymyoglobin.

Agreed. In the Methods we had mentioned that all CO solutions were prepared by bubbling bath solution with CO gas; this creates a saturated CO solution ([CO] = 1mM) as we had mentioned in the kinetics section of the Methods. However, to make it clearer for the reader we have adjusted the electrophysiology part of the Online methods section so that it now reads: "All CO solutions were prepared by bubbling bath solution with CO gas; this creates a saturated CO solution ([CO] = 1mM³). The difficulty of these experiments makes it unfeasible to carry out a CO concentration dependent study - our objective was to reliably confirm or rule out a heme-dependent CO effect on the K_{ATP} channel.

The physiological role of this interaction has not been demonstrated, the studies have focused on overexpression of channel protein. How do the authors perceive that this interaction occurs at a system level?

The purpose of this paper was to quantify (at the protein level) the mechanism of heme and CO binding in K_{ATP} channels; the physiological consequences are well beyond the scope of this work. We can speculate on the physiological effects, so a sentence has been added in the second paragraph from the end of the discussion to highlight potential physiological effects.

Mutational work (Figure 1b); are all residues required for the observed CO effect?

This is addressed in the reply to reviewer 1's comments on the heme mutations.

Functional datasets (Figure 1b); single channel K+ conductance plots should be presented.

Since the recordings were made at equimolar (140 mM) K⁺ the single channel conductance would be virtually constant over a wide range of membrane potentials as predicted by the GHK equation. A

sentence has been added to the methods section: 'The conductance of the K_{ATP} channel was ~ 78 pS under our conditions which is consistent with values observed for Kir6.2 and SUR2A channel recordings.'

Why did the authors select +70mV as a test potential?

A test potential of -70mV was selected as this ensures a high driving force at a voltage where other channels are not activated. A sentence in the methods has been included to explain this.

Did the authors estimate the number of channels per patch? What was the sample size for the data presented?

A note has been added to the methods giving an indication of the number of channels per patch ('The number of channels per patch was variable with an average of 5.4 ± 0.3 channels per patch.')

Minor Comments

P2. 'Inside-out patches were superfused with heme and an increase in Popen was observed.' The data does not support this statement and it should be revised.

This issue has now been resolved and the revised 2nd paragraph of the results. We hope this is acceptable to the reviewer.

Reviewers' Comments:

Reviewer #1:

Remarks to the Author:

I think that, with two relatively minor modifications related to the original review, this paper is ready for publication.

4) Line 159: Given that there is no overlap, even at the extremes of the error bars ($0.19+0.1=0.29 \mu\text{M}$ vs $0.60-0.30=0.30 \mu\text{M}$), I suggest that "...kinetically, is in **reasonably** close agreement..." might be a better description. The inserted word is in bold.

5) Lines 225-226: It would be more accurate to state that a distribution of enthalpic barriers is needed to account for the temperature dependence of the kinetics. In addition, the word "wide" would more properly be associated with the enthalpic barrier distribution, rather than with the underlying coordinate or conformational distribution. I suggest this because a conformational substate distribution, which generates only a sub-Å width ($\sim 0.1 \text{ \AA}$) in the heme doming equilibrium position, can quantitatively account for the non-exponential kinetics over a very wide temperature range. Under slow conformational relaxation conditions, this relatively narrow distribution of equilibrium positions leads to a barrier distribution that is relatively broad, or "wide", when using a doming force constant that correlates with the observed low frequency doming motion.

A more technically correct phrasing of lines 225-226 might be (changes in bold): "This large span of rebinding rates is likely to reflect a wide **enthalpic barrier distribution due to different** configurations of the dissociated heme and its environment (as assigned by Champion and coworkers in very recent work on the temperature dependence of heme-CO rebinding in CooA33 and in free heme in solution³⁴). **Orientational** configurations of dissociated CO in the heme pocket **may also contribute to the kinetic response**³⁵. "

Reviewer #2:

None

Reviewer #3:

Remarks to the Author:

The authors have adequately addressed my concerns as raised in the original review. Therefore, I am happy to recommend publication of this manuscript.